# Neurophysiological Markers of Statistical Learning in Music and Language: Hierarchy, Entropy and Uncertainty

**DOI:** 10.3390/brainsci8060114

**Published:** 2018-06-19

**Authors:** Tatsuya Daikoku

**Affiliations:** Department of Neuropsychology, Max Planck Institute for Human Cognitive and Brain Sciences, 04103 Leipzig, Germany; daikoku@cbs.mpg.de; Tel.: +81-5052157012

**Keywords:** statistical learning, implicit learning, domain generality, information theory, entropy, uncertainty, order, *n*-gram, Markov model, word segmentation

## Abstract

Statistical learning (SL) is a method of learning based on the transitional probabilities embedded in sequential phenomena such as music and language. It has been considered an implicit and domain-general mechanism that is innate in the human brain and that functions independently of intention to learn and awareness of what has been learned. SL is an interdisciplinary notion that incorporates information technology, artificial intelligence, musicology, and linguistics, as well as psychology and neuroscience. A body of recent study has suggested that SL can be reflected in neurophysiological responses based on the framework of information theory. This paper reviews a range of work on SL in adults and children that suggests overlapping and independent neural correlations in music and language, and that indicates disability of SL. Furthermore, this article discusses the relationships between the order of transitional probabilities (TPs) (i.e., hierarchy of local statistics) and entropy (i.e., global statistics) regarding SL strategies in human’s brains; claims importance of information-theoretical approaches to understand domain-general, higher-order, and global SL covering both real-world music and language; and proposes promising approaches for the application of therapy and pedagogy from various perspectives of psychology, neuroscience, computational studies, musicology, and linguistics.

## 1. Introduction

The brain is a learning system that adapts to multiple external phenomena existing in its living environment, including various types of input such as auditory, visual, and somatosensory stimuli, and various learning domains such as music and language. By means of this wide-ranging system, humans can comprehend structured information, express their own emotions, and communicate with other people [1]. According to linguistic [2,3] and musicological studies [4,5], music and language have domain-specific structures including universal grammar, tonal pitch spaces, and hierarchical tension. Neurophysiological studies likewise suggest that there are specific neural bases for language [6,7] and music comprehension [8,9]. Nevertheless, a body of research suggests that the brain also possesses a domain-general learning system, called statistical learning (SL), that is partially shared by music and language [10,11]. SL is a process by which the brain automatically calculates the transitional probabilities (TPs) of sequential phenomena such as music and language, grasps information dynamics without an intention to learn or awareness of what we know [12,13], and further continually updates the acquired statistical knowledge to adapt to the variable phenomena in our living environments [14]. Some researchers also indicate that the sensitivity to statistical regularities in sequences could be a by-product of chunking [15].

The SL phenomenon can partially be supported by a unified brain theory [16]. This theory tries to provide a unified account of action and perception, as well as learning under a free-energy principle [17,18], which views several keys of brain theories in the biological (e.g., neural Darwinism), physical (e.g., information theory), and neurophysiological (e.g., predictive coding) sciences. This suggests that several brain theories might be unified within a free-energy framework [19], although its capacity to unify different perspectives has yet to be established. This theory suggests that the brain models phenomena in its living environment as a hierarchy of dynamical systems that encode a causal chain structure in the sensorium to maintain low entropy [16], and predicts a future state based on the internalized model to minimize sensory reaction and optimize motor action. This prediction is in keeping with the theory of SL in the brain. That is, in SL theory, the brain models sequential phenomena based on TP distributions, grasps entropy in the whole sequences, and predicts a future state based on the internalized stochastic model in the framework of predictive coding [20] and information theory [21]. The SL also occurs in action sequences [22,23], suggesting that SL could contribute to optimization of motor action.

SL is considered an implicit and ubiquitous process that is innate in humans, yet not unique to humans, as it is also found in monkeys [24,25], songbirds [26,27], and rats [28]. The terms implicit learning and SL have been used interchangeably and are regarded as the same phenomenon [15]. A neurophysiological study [29] has suggested that conditional probabilities in the Western music corpus are reflected in the music-specific neural responses referred to as early right anterior negativity (ERAN) in event-related potential (ERP) [8,9]. The corpus study also found statistical universals in music structures across cultures [30,31]. These findings also suggest that musical knowledge may be at least partially acquired through SL. Our recent studies have also demonstrated that the brain codes the statistics of auditory sequences as relative information, such as relative distribution of pitch and formant frequencies, and that this information can be used in the comprehension of other sequential structures [10,32]. This suggests that the brain does not have to code and accumulate all received information, and thus saves some memory capacity [33]. Thus, from the perspective of information theory [21], the brain’s SL is systematically efficient.

As a result of the implicit nature of SL, however, humans cannot verbalize exactly what they statistically learn. Nonetheless, a body of evidence indicates that neurophysiological and behavioural responses can unveil musical and linguistic SL effects [14,32,34,35,36,37,38,39,40,41,42,43,44] in the framework of predictive coding [20]. Furthermore, recent studies have detected the effects of musical training on linguistic SL of words [41,43,45,46,47] and the interactions between musical and linguistic SL [10] and between auditory and visual SL [44,48,49,50]. On the other hand, some studies have also suggested that SL is impaired in humans with domain-specific disorders such as dyslexia [51,52,53] and amusia [54,55], disorders that affect linguistic and music processing, respectively (though Omigie and Stewart (2011) [56] have suggested that SL is intact in congenital amusia). Thiessen et al. [57] suggested that a complete-understanding statistical learning must incorporate two interdependent processes: one is the extracting process that computes TPs (i.e., local statistics) and extracts each item, such as word segmentation, and the other one is the integration process that computes distributional information (i.e., summary statistics) and integrates information across the extracted items. The entropy and uncertainty (i.e., summary statistics), as well as TPs, are used to understand the general predictability of sequences in domain-general SL that could cover music and language in the interdisciplinary realms of neuroscience, behavioral science, modeling, mathematics, and artificial intelligence. Recent studies have suggested that SL strategies in the brain depend on the hierarchy, order [14,35,58,59], entropy, and uncertainty in statistical structures [60]. Hasson et al. [61] also indicated that certain regions or networks perform specific computations of global or summary statistics (i.e., entropy), which are independent of local statistics (i.e., TP). Furthermore, neurophysiological studies suggested that sequences with higher entropy were learned based on higher-order TP, whereas those with lower entropy were learned based on lower-order TP [59]. Thus, it is considered that information-theoretical and neurophysiological concepts on SL link each other [62,63]. The integrated approach of neurophysiology and informatics based on the notion of order of TP and entropy can shed light on linking concepts of SL among a broad range of disciplines. Although there have been a number of studies on SL in music and language, few studies have examined the relationships between the “order” of TPs (i.e., the order of local statistics) and entropy (i.e., summary statistics) in SL. This article focuses on three themes in SL from the viewpoint of information theory, as well as neuroscience: (1) a mathematical interpretation of SL that can cover music and language and the experimental paradigms that have been used to verify SL; (2) the neural basis underlying SL in adults and children; and (3) the applicability of therapy and pedagogy for humans with learning disabilities and healthy humans.

## 2. Mathematical Interpretation of Brain SL Process Shared by Music and Language

### 2.1. Local Statistics: Nth-Order Transitional Probability

According to SL theory, the brain automatically computes TP distributions in sequential phenomena (local statistics) [35], grasps uncertainty/entropy in the whole sequences (global statistics) [61], and predicts a future state based on the internalized statistical model to minimize sensory reaction [16,20]. The TP is a conditional probability of an event B given that the latest event A has occurred, written as P(B|A). The TP distributions sampled from sequential information such as music and language are often expressed by *n*th-order Markov models [64] or *n*-gram models [21] (Figure 1). Although the terminology of *n*-gram models has frequently been used in natural language processing, it has also recently been used in music models [65,66]. They have often been applied to develop artificial intelligence that gives computers learning abilities similar to those of the human brain, thus generating systems for data mining, automatic music composition [67,68,69], and automatic text classification in natural language processing [70,71]. The mathematical model of SL including *n*th-order Markov and (*n* + 1)-gram models is the conditional probability of an event *e_n_*_+1_, given the preceding n events based on Bayes’ theorem:*P*(*e_n_*_+1_|*e_n_*) = *P*(*e_n_*_+1_ ∩ *e_n_*)/*P*(*e_n_*)(1)

From the viewpoint of psychology, the formula can be interpreted as positing that the brain predicts a subsequent event *e_n_*_+1_ based on the preceding events *e_n_* in a sequence. In other words, learners expect the event with the highest TP based on the latest n states, whereas they are likely to be surprised by an event with lower TP (Figure 2).

### 2.2. Global Statistics: Entropy and Uncertainty

SL models are sometimes evaluated in terms of entropy [72,73,74,75] in the framework of information theory, as done by Shannon [21]. Entropy can be calculated from probability distribution, interpreted as the average surprise (uncertainty) of outcomes [16,76], and used to evaluate the neurobiology of SL [60], as well as rule learning [77], decision making [78], anxiety, and curiosity [79,80] from the perspective of uncertainty. For instance, the conditional entropy (H(B|A)) in the *n*th order TP distribution (hereafter, Markov entropy) can be calculated from information contents:*H(X*_*i*+1_|*X_i_)* = −*ΣP(x_i_)ΣP(x*_*i*+1_|*x_i_)**log*_2_*P(x*_*i*+1_|*x_i_)*(2)
where *H*(*X_i_*_+1_|*X_i_*) is the Markov entropy; *P*(*X_i_*) is the probability of event xi occurring; and *P*(*X_i_*_+1_|*X_i_*) is the probability of *X_i_*_+1_, given that *X_i_* occurs previously. Previous articles have suggested that the degree of Markov entropy modulates human predictability in SL [61,81]. The uncertainty (i.e., global/summary statistics), as well as the TP (i.e., local statistics), of each event is applicable to and may be used to predict many types of sequential distributions, such as music and language, and to understand the predictability of a sequence (Figure 3). Indeed, entropy and uncertainty are often used to understand domain-general SL in the interdisciplinary realms of neuroscience, behavioural science, modeling, mathematics, and artificial intelligence.

### 2.3. Experimental Designs of SL in Neurophysiological Studies

The word segmentation paradigm is frequently used to examine the neural basis underlying SL (e.g., [34,41,43,44,46,82,83,84,85,86,87,88,89,90,91,92,93,94,95,96]). This paradigm basically consists of a concatenation of pseudo-words (Figure 2a). In the pseudo-words sequence, the TP distributions based on a first-order Markov model represent lower TPs in the “first” stimulus of each word (Figure 2a: P(B|A), P(C|B), and P(A|C)) than other stimuli of word (Figure 2a: P(C|A), P(A|A), P(A|B), P(B|B), P(B|C), and P(C|C)). When the brain statistically learns the sequences, it can identify the boundaries between words based on first-order TPs (Figure 2a) [97,98], and segment/extract each word. The SL of word segmentation based on first-order TPs has been considered as a mechanism for language acquisition in the early stages of language learning, even in infancy [12]. Recent studies have also demonstrated that SL can be performed based on within-word, as well as between-word, TPs ([40,98] for example, see Figure 2d). Although a number of studies have used a word segmentation paradigm consisting of words with a regular unit length (typically, three stimuli within a word), previous studies suggest that the unit length of words [99], the order of TPs [59], and the nonadjacent dependencies of TPs in sequences ([14,100,101,102] for example, see Figure 2c) can modulate the SL strategy used by the brain. Indeed, natural languages and music make use of higher-order statistics, including hierarchical, syntactical structures. To understand the brain’s higher-order SL systems in a form closer to that used for natural language and music, sequential paradigms based on higher-order Markov models have also been used in neurophysiological studies ([32,35,103] for example, see Figure 2b). Furthermore, the *n*th-order Markov model has been applied to develop artificial intelligence that gives computers learning and decision-making abilities similar to those of the human brain, thus generating systems for automatic music composition [67,68,69] and natural language processing [70,71]. Information-theoretical approaches, including information content and entropy based on *n*th-order Markov models, may be useful in understanding the domain-general SL, as it functions in response to real-world learning phenomena in the interdisciplinary realms of brain and computational sciences.

## 3. Neural Basis of Statistical Learning

### 3.1. Event-Related Responses and Oscillatory Activity

The ERP and event-related magnetic fields (ERF) modalities directly measure brain activity during SL and represent a more sensitive method than the observation of behavioral effects [40,41,104]. Based on predictive coding [20], when the brain encodes the TP distributions of a stimulus sequence, it expects a probable future stimulus with a high TP and inhibits the neural response to predictable external stimuli for efficiency of neural processing. Finally, the effects of SL manifest as a difference in the ERP and ERF amplitudes between stimuli with lower and higher TPs (Figure 4). Although many studies of word segmentation detected SL effects on the N400 component [43,46,88,89,93,94,105], which is generally considered to reflect a semantic meaning in language and music [106,107,108], auditory brainstem response (ABR) [96], P50 [41], N100 [94], mismatch negativity (MMN) [40,44,98], P200 [46,89,105], N200–250 [44,47], and P300 [83] have also been reported to reflect SL effects (Table 1). In addition, other studies using Markov models also reported that SL is reflected in the P50 [14,36,37], N100 [10,14,32,35], and P200 components [35]. Compared with later auditory responses such as N400, the auditory responses that peak earlier than 10 ms after stimulus presentation (e.g., ABR) and at 20–80 ms, which is around P50 latency, have been attributed to parallel thalamo–cortical connections or cortico–cortical connections between the primary auditory cortex and the superior temporal gyrus [109]. Thus, the suppression of an early component of auditory responses to stimuli with a higher TP in lower cortical areas can be interpreted as the transient expression of prediction error that is suppressed by predictions from higher cortical areas in a top-down connection [96]. Thus, top-down, as well as bottom-up, processing in SL may be reflected in ERP/ERF. On the other hand, SL effects on N400 have been detected in word-segmentation tasks, but not in the Markov model. TPs of a word-segmentation task are calculated based on first-order models (Figure 2a). In other words, in terms of the “order” of TP, SL of word segmentation (i.e., sequence consisting of word concatenation) and first-order Markov model have same hierarchy of TP. Nevertheless, SL studies using the first-order Markov model did not detect learning effects of N400 (Table 1). The phenomenon of word segmentation itself has been considered as a mechanism of language acquisition in the early stages of language learning [12]. Several papers claim that the sensitivity to statistical regularities in sequences of word concatenation could be a by-product of chunking [15]. Neurophysiological effects of word segmentation, such as N400, reflecting a semantic meaning in language [106,107,108] may be associated with the neural basis underlying linguistic functions, as well as statistical computation itself. On the other hand, our previous study using the first-order Markov model [36] struggled to detect N400 in terms of a stimulus onset asynchrony of sequences (i.e., 500 ms). A future study will be needed to verify SL effects of N400 using the Markov model.

It has been suggested that SL could also be reflected in oscillatory responses in the theta band [115,116]. Moreover, the human and monkey auditory cortices represent the neural marker of predictability based on SL in the form of modulations of transient theta oscillations coupling with gamma and concomitant effects [25], suggesting that SL processes are unlikely to have evolved convergently and are not unique to humans. According to previous studies, low-frequency oscillations may play an important role in speech segmentation associated with SL [73], and in tracking the envelope of the speech signal, whereas high-frequency oscillations are fundamentally involved in tracking the fine structure of speech [117]. Furthermore, there is evidence of top-down effects in low-frequency oscillations during listening to speech (up to beta band: 15–30 Hz), whereas bottom-up processing dominates in higher frequency bands [118]. Studies on the auditory oddball paradigm have also demonstrated that the power and/or coherence of theta oscillations to low-probability sounds is increased relative to high-probability sounds. Thus, many studies suggest that the lower-frequency oscillations, including theta band, are related to the prediction error [119]. Top-down predictions also control the coupling between speech and low-frequency oscillations in the left frontal areas, most likely in the speech motor cortex [120]. Although low-frequency oscillations could cover ERP components that have been suggested to reflect SL effects, the studies on oscillation and prediction imply the importance of investigating SL effects on oscillatory responses, as well as ERP.

### 3.2. Anatomical Mechanisms

#### 3.2.1. Local Statistics: Transitional Probability

Neuroimaging studies have indicated that both cortical and subcortical areas play an important role in SL. For instance, the auditory association cortex, including the superior temporal sulcus (STS) [91] and superior temporal gyrus (STG) [110], contributes to auditory SL of both speech and non-speech sounds. Previous studies have also reported the effects of laterality on SL. For instance, functional magnetic resonance imaging (fMRI) [121] and near-infrared spectroscopy (NIRS) [111] studies have suggested that SL is linked to the left auditory association cortex or the left inferior frontal gyrus (IFG) [112,122], which include Wernicke’s and Broca’s areas, respectively. Furthermore, one previous study has indicated that brain connectivity between bilateral superior temporal sources and the left IFG is important for auditory SL [45]. On the other hand, another study has shown that the right posterior temporal cortex (PTC), which represents the high levels of the peri-Sylvian auditory hierarchy, is related to higher-order auditory SL [35] (i.e., second-order TPs). Further study will be needed to examine the relationships between the order of TPs in sequences and the neural correlations that depend on the order of TPs and hierarchy of SL.

Some studies have suggested that the sensory type of each stimulus modulates the neural basis underlying SL. For instance, some previous studies have suggested that the right hemisphere contributes to visual SL [123]. Paraskevopoulos and colleagues [50] revealed that the cortical network underlying audiovisual SL was partly common with and partly distinct from the unimodal networks of visual and auditory SL, comprising the right temporal and left inferior frontal sources, respectively. fMRI studies have also reported that Heschl’s gyrus and the medial temporal lobe [124] contribute to auditory and visual SL, respectively [113], and that motor cortex activity also contributes to visual SL of action words [22]. Furthermore, Cunillera et al. [88] have suggested that the superior part of the ventral premotor cortex (PMC), as well as the posterior STG, are responsible for SL of word segmentation, suggesting that linguistic SL is related to an auditory–motor interface. Another study has suggested that the abstraction of acquired statistical knowledge is associated with a gradual shift from memory systems in the medial temporal lobe, including the hippocampus, to those of the striatum, and that this may be mediated by slow wave sleep [125].

#### 3.2.2. Global Statistics: Entropy

Perceptive mechanisms of summary structure (i.e., global statistics) are considered to be independent of the prediction of each stimulus with different TPs (local statistics) [57,61]. Recent studies have examined the brain systems that are responsible for encoding the uncertainty of global statistics in sequences by comparing brain activities while listening to Markov/word-concatenation and random sequences, which have lower and higher entropies, respectively. Regardless of whether music or language is assessed, the hippocampus and the lateral temporal region [88], including Wernicke’s area [114], are considered to play important roles in encoding uncertainty and conditional entropy of statistical information [60]. Bischoff-Grethe et al. have also indicated that Wernicke’s area may not be exclusively associated with uncertainty of language information [114]. Furthermore, uncertainty in auditory and visual statistics is coded by modality-general, as well as modality-specific, neural mechanisms [126,127], supporting the hypothesis that the neural basis underlying the brain’s perception of global statistics (i.e., uncertainty), as well as local statistics (i.e., prediction of each stimulus with different TPs), is a domain-general system. Our previous neural study also suggested that reorganization of acquired statistical knowledge requires more time than the acquisition of new statistical knowledge, even if the new and previously acquired information sets have equivalent entropy levels [14]. Furthermore the results suggested that humans learn larger structures, such as phrases, first and subsequently extract smaller structures, such as words, from the learned phrases (global-to-local learning strategy). To the best of our knowledge, however, no study has yet demonstrated the differences and neural basis interactions between global and local statistics. Further study is needed to reveal how the coding of global statistics affects that of local statistics.

## 4. Clinical and Pedagogical Viewpoints

### 4.1. Disability

Although SL is a domain-general system, some studies have reported that SL is impaired in domain-specific disabilities such as dyslexia [51,52,53] and amusia [54,55], which are language- and music-related disabilities, respectively. Ayotte and colleagues [128] have suggested that individuals with congenital amusia fail to learn music SL but can learn linguistic SL, even if the sequences of both types have the same degree of statistical regularity [54]. Another study has suggested, in contrast, that SL is intact in amusia [56], and that individuals with amusia lack confidence in their SL ability, although they can engage in SL of music. Peretz et al. [54] stated that the input and output of the statistical computation might be domain-specific, whereas the learning mechanism might be domain-general. Furthermore, previous studies have indicated that SL ability is impaired in patients with damage to a specific area of the brain. For instance, SL is impaired in connection with hippocampal [129] and right-hemisphere damage [130]. Indeed, it has been suggested that the hippocampus plays an important role in SL [124]. One recent study indicated that auditory deprivation leads to disability of not only auditory SL [131] but also visual SL [132]. This implies that there may be specific neural mechanisms for SL that can be shared among distinct sensory modalities. Another study [133], however, suggested that a period of early deafness is not associated with SL disability. Further study is needed to clarify whether SL disability is related to temporary auditory deprivation.

### 4.2. Music-to-Language Transfer

#### 4.2.1. Neural Underpinnings of SL That Overlap across Music and Language Processing

Because of the acoustic similarity [134], cortical overlap [135,136], and domain generality of SL across language and music, experienced listeners to particular spectrotemporal acoustic features, such as rhythm and pitch, in either speech or music have an advantage when perceiving similar features in the other domain [137]. According to neural studies, musical training leads to a different gray matter concentration in the auditory cortex [138] and a larger planum temporale (PT) [139,140,141,142,143]; the region where both language and music are processed. An ERP study has demonstrated that both the linguistic and the musical effects of SL on the N100–P200 response, which could originate in the belt and parabelt auditory regions [144,145], were larger in musicians than in non-musicians [46]. Thus, the increased PT volume associated with musical training may facilitate auditory processing in SL. A magnetoencephalographic (MEG) study also reported that the effect of SL on the P50 response was larger in musicians than in non-musicians [41], suggesting that musical training also boosts corticofugal projections in a top-down manner regarding predictive coding [96].

Musical training could also facilitate the effects of SL on N400 [46], which is considered to be associated with IFG and PMC [88]. According to the results of a neural study, musicians have an increased gray matter density of the left IFG (i.e., Broca’s area) and PMC [146]. Other studies have suggested that, during SL of word segmentation, musicians exhibit increased left-hemispheric theta coherence in the dorsal stream projecting from the posterior superior temporal (pST) and inferior parietal (IP) brain regions toward the prefrontal cortex, whereas non-musicians show stronger functional connectivity in the right hemisphere [115]. An MRI study also demonstrated that SL of word segmentation leads to pronounced left-hemisphere activity of the supratemporal plane, IP lobe, and Broca’s area [147]. Thus, the left dorsal stream is considered to play an important role in SL, as well as language [7] and music learning [148].

The SL of word segmentation plays an important role in various speech abilities. Recent studies have revealed a strong link between SL of word segmentation and more general linguistic proficiency such as expressive vocabulary [149] and foreign language [150]. An fMRI study [151] has suggested that, during SL of word segmentation, participants with strong SL effects of familiar language on which they had been pretrained had decreased recruitment of fronto-subcortical and posterior parietal regions, as well as a dissociation between downstream regions and early auditory cortex, whereas participants with strong SL effects of novel language that had never been exposed showed the opposite trend. Furthermore, children with language disorders perform poorly when compared with typical developing children in tasks involving musical metrical structures [152], and have more difficulty in SL of word segmentation [153] and perception of speech rhythms [154,155]. Thus, musical training, including rhythm perception and production, is important for the development of language skills in children. Together, a body of study indicates that musical expertise may transfer to language learning [104]. It is generally considered that the left auditory cortex is more sensitive to temporal information, such as musical beat and the voice-onset (VOT) time of consonant-vowel (CV) syllables, whereas the right auditory cortex plays a role in spectral perception, such as pitch and vowel discriminations. Recent studies have indicated relationships between rhythm perception and SL [156].

Recent neural studies have demonstrated that SL of speech, pitch, timbre, and chord sequences can be performed and reflected in ERP/ERF [10,36,37,40,46]. Furthermore, the brain codes statistics of auditory sequences as relative information, such as relative distribution of pitch and formant frequencies, which could be used for comprehension of another sequential structure [10,32], suggesting that SL is ubiquitous and domain-general. On the other hand, the relative importance of acoustic features such as rhythm, pitch, intensity, and timbre varies depending on the domain, that is, music or language [157]. For instance, unlike spoken language, music contains various pitch frequencies. Recent studies have suggested that, compared with speech sequences, sung sequences with various pitches facilitate auditory SL based on word segmentation [92] and the Markov model [10]. These results further support the advantage of musical training for language SL. In addition, Hansen and colleagues have suggested that musical training also facilitates the hippocampal perception of global statistics of entropy (i.e., uncertainty) [158], as well as local statistics of each TP. Thus, musical training contributes to the improvement of SL systems in various brain regions, including the auditory cortex. Together, the facilitation of SL may be related to enhancement of the left dorsal stream via the IFG and PMC, as well as PT, enhanced low-level auditory processing in a top-down manner, and enhanced hippocampal processing. Musical training including rhythm perception contributes to these enhancements and facilitates the involvement of SL in language skills, and thus could be an important clinical and pedagogical strategy in persons with any of a variety of language-related disorders such as dyslexia [159,160] and aphasia [161].

#### 4.2.2. Children and Adults: Critical Periods and Plasticity in the Brain

Previous studies have demonstrated that auditory SL can be performed even by sleeping neonates [85,86,162]. SL is ubiquitously performed at birth, showing that the human brain is innately prepared for it. An infant’s SL extends to rhythms [163], visual stimuli [164], objects [165], social learning [23,166], and a general mechanism by which infants form meaningful representations of the environment [167]. Furthermore, infants can also learn non-adjacent statistics [101]. This suggests that SL plays an important role in an infant’s syntactic learning, as well as the simple segmentation of words. These results may enable us to disentangle the respective contributions of nature and nurture in the acquisition of language and music. On the other hand, an MEG study has suggested that the strategies for language acquisition in infants could shift from domain-general SL to domain-specific processing of native language between 6 and 12 months [116], a “critical period” for language acquisition [168]. A comparable developmental change from domain-general to domain-specific learning strategies can also occur in music perception [169]. During the “critical period” of heightened plasticity, the brain is formed by sensory experience [170,171,172]. The development of primary cortical acoustic representations can be shaped by the higher-order TP of stimulus sequences [58]. An ERP study [173] suggested that sensitivity to speech stimuli in infants gradually shifts from accentuation to repetition during a critical period. These results may suggest that cortical reorganization depending on early experience interacts with SL [174], and that fluctuations in the degree of dependence on SL for the acquisition of language and music are part of the developmental process during critical periods. On the other hand, the SL system in the brain can be preserved even in adults (e.g., [32,35,40,41]). According to previous studies, neural plasticity can occur in adults through SL [175] and musical training [176]. In fact, there is no doubt that SL occurs in adults who are already beyond the critical periods, and that their SL ability can be modulated by auditory training. Recent studies have revealed that the process of reorganization of acquired statistical knowledge can be detected in neurophysiological responses [14]. Furthermore, a computational study on music suggested the possibility that the time-course variation of statistical knowledge over a composer’s lifetime can be reflected in that composer’s music from different life stages [177]. Thus, implicit updates of statistical knowledge could be enabled by the combined and interdisciplinary approach of brain, behavioral, and computational methodologies [178].

## 5. General Discussion

### 5.1. Information-Theoretical Notions for Domain-General SL: Order of TP and Entropy

SL is a domain-general and interdisciplinary notion in psychology, neuroscience, musicology, linguistics, information technology, and artificial intelligence. To generate SL models that are applicable to all of these various realms, the *n*th-order Markov and *n*-gram models based on information theory have frequently been used in natural language processing [70,71] and in the creation of automatic music composition systems [67,68,69]. Such models can verify hierarchies of SL based on various-order TPs. Natural languages and music include higher-order statistics, such as hierarchical syntactical structures and grammar. Thus, information-theoretical approaches, including information content and entropy based on *n*th-order Markov models [59,61,81], can express domain-general statistical structures closer to those of real-world language and music. The SL models are often evaluated in terms of entropy [72,73,74,75]. From a psychological viewpoint, entropy is interpreted as the average surprise (uncertainty) of outcomes [16,76]. Previous studies have demonstrated that the perception of entropy and uncertainty based on SL could be reflected in neurophysiological responses [59] and activity of the hippocampus [60]. Hasson et al. [61] indicated that certain regions or networks perform specific computations of global or summary statistics (i.e., entropy), which are independent of local statistics (i.e., TP). Furthermore, Thiessen and colleagues [57] proposed that a complete-understanding statistical learning must incorporate two interdependent processes: one is the extracting process that computes TPs and extracts each item, such as word segmentation, and the other one is the integration process that computes distributional information and integrates information across the extracted items. Our previous studies [59] investigated correlation among entropy, order of TP, and the SL effect. As a result, the SL effects of sequences with higher entropy were lower than those with lower entropy, even when TP itself is same between these two sequences. This suggests that an evaluation of computational model of sequential information by entropy in the field of informatics may partially be able to predict learning effect in human’s brain. Thus, the integrated methodology of neurophysiology and informatics based on the notion of entropy can shed light on linking the concept of SL among a broad range of disciplines. To understand the domain-general SL system that incorporates notions from both information theory and neuroscience, it is important to investigate both global and local SL.

### 5.2. Output of Statistical Knowledge: From Learning to Using

According to recent studies, acquired statistical knowledge contributes to the comprehension and production of complex structural information, such as music and language [179], intuitive decision-making [77,78,180,181,182], auditory-motor planning [183], and creativity involved in musical composition [62]. Several studies suggest that musical representation is mainly formed by a tacit knowledge [184,185,186]. Thus, statistical knowledge is closely tied to musical and speech expression such as composition, playing, and conversation. In addition, global statistical knowledge (i.e., entropy and uncertainty), as well as local statistical knowledge (each TP), is also supposed to contribute to decision-making [78], anxiety [80], and curiosity [79]. A number of studies have reported, however, that humans cannot verbalize exactly what they have learned statistically, even when an SL effect is detected in neurophysiological responses [14,32,34,35,36,37,38,39,40,41,42,43,44]. Nevertheless, our previous study suggested that statistical knowledge could alternatively be expressed via abstract medium such as musical melody [32]. In these studies, learners could behaviorally distinguish between sequences with more than eight tones with only higher TPs and those with only lower TPs, suggesting that humans can distinguish sequences with different TPs when they are provided longer sequences when compared with a conventional way in word-segmentation studies that present sequences with three tones. These studies may also suggest that that SL of auditory sequences partially interact with the Gestalt principle [5]. Furthermore, an fMRI study has suggested that the abstraction of statistical knowledge is associated with a gradual shift from the memory systems in the medial temporal lobe, including the hippocampus, to those of the striatum, and that this may be mediated by slow wave sleep [125]. Future study is needed to examine how/when statistical learning contributes to mental expression of music and language.

### 5.3. Applicability in Clinical and Pedagogy

Previous studies suggest that neurophysiological correlations of SL can disclose subtle individual differences that might be underestimated by behavioral levels [34,88,89,187], although recent studies showed individual differences in SL by behavioral tasks [188]. Some studies suggest that neurophysiological responses disclose SL effects, even when no SL effects cannot be detected in behavioral levels [40,41]. Neurophysiological markers of SL may at least be informative when studying less accessible populations such as infants, who are unable to deliver an obvious behavioral response [86,162]. For instance, ERP/ERF could be a useful method for the evaluation of the individual ability of SL, which is linked to individual skill in language and music learning [189,190], and which is impaired in humans with language- and music-based learning impairments such as dyslexia [51,52,53] and amusia [54,55]. Thus, neurophysiological markers of SL may be applicable for the evaluation of therapeutic and educational effects for patients and healthy humans [191] across any domain in which the conditional probabilities of sequential events vary systematically. Francois’s findings [43] suggest the possibility of music-based remediation for children with language-based SL impairments. In addition, by using information theoretic approaches such as higher-order Markov models and entropy, SL ability can be evaluated in the form that is closest to that used in learning natural language and music [14,63]. The integration of neural, behavioral, and information-theoretical approaches may enhance our ability to evaluate SL ability in terms of both music and language.

### 5.4. Challenges and Future Prospects: SL in Real-World Music and Language

Although SL is generally considered domain-general, many studies also report that comprehension of language and music, which have domain-specific structures including universal grammar, tonal pitch spaces, and hierarchical tension [2,3,4,5], may rely on domain-specific neural bases [6,7,8,9,192]. Furthermore, current SL paradigms are not sufficient to account for all levels of the music- and language-learning process. Some studies suggest two steps of the learning process [193,194]. The first is SL, which shares a common mechanism among all the domains (domain generality). The second is domain-specific learning, which has different mechanisms in each domain (domain specificity). This learning process implies that, at least in an earlier step of the learning process, SL plays an essential role that covers music and language learning abilities [195]. On the other hand, few studies investigated how statistically acquired knowledge was represented in real-world communication, conversation, action, and music expression. Future studies will be needed to investigate how neural systems underlying SL contribute to comprehension and production in real-world music and language. Information-theoretical approaches based on higher-order Markov models can be used to understand SL systems in a form closer to that used for natural language and music, from a perspective of linguistics, musicology, and a unified brain theory such as the free-energy principle [16], including optimisation of action, as well as perception and learning.

## 6. Conclusions

This paper reviews a body of recent neural studies on SL in music and language, and discusses the possibility of therapeutic and pedagogical application. Because of a certain degree of acoustic similarity, neural overlap, and domain generality of SL between speech and music, musical training positively affects language skills in SL. Recent studies also suggested that SL strategies in the brain depend on the hierarchy, order [14,35,58,59], entropy, and uncertainty in statistical structures [60], and that certain brain regions perform specific computations of entropy that are independent of those of TP [61]. Yet few studies have investigated the relationships between the order of TPs (i.e., order of local statistics) and entropy (i.e., global statistics) in terms of SL strategies of the human brain. Information-theoretical approaches based on higher-order Markov models that can express hierarchical information dynamics as they are expressed in real-world language and music represent a possible means of understanding domain-general, higher-order, and global SL in the interdisciplinary realms of psychology, neuroscience, computational studies, musicology, and linguistics.

## Figures and Tables

**Figure 1 brainsci-08-00114-f001:**
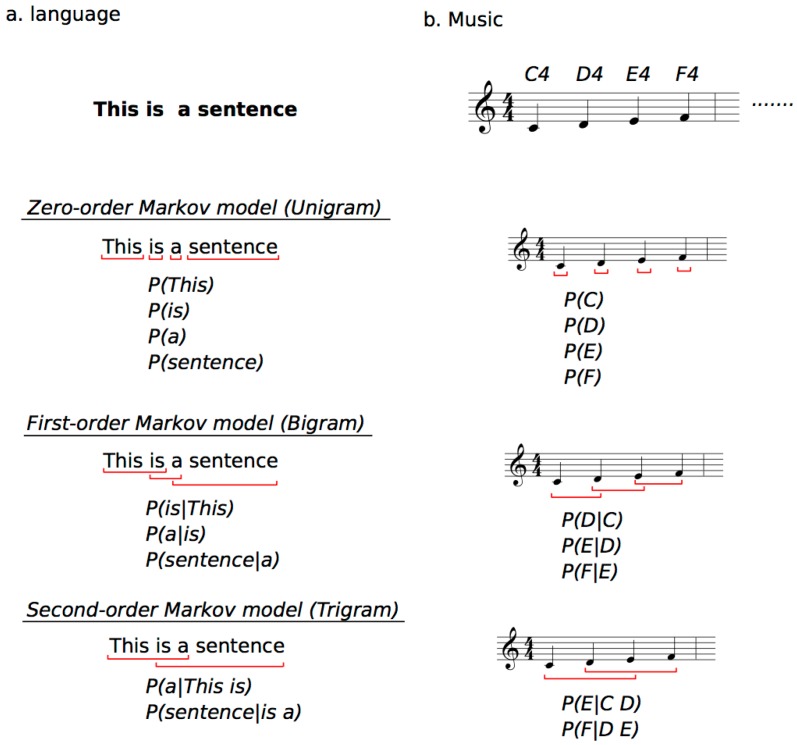
Example of *n*-gram and Markov models in statistical learning (SL) of language (**a**) and music (**b**) based on information theory. The top are examples of sequences, and the others explain how to calculate TPs (*P*(*e_n_*_+1_|*e_n_*)) based on zero- to second-order Markov models. They are based on the conditional probability of an event *e_n_*_+1_, given the preceding n events based on Bayes’ theorem. For instance, in language ((**a**), This is a sentence), the second-order Markov model represents that the “a” can be predicted based on the last subsequent two words of “This” and “is”. In music ((**b**), C4, D4, E4, F4), second-order Markov model represents that the “E” can be predicted based on the last subsequent two tones of “C” and “D”.

**Figure 2 brainsci-08-00114-f002:**
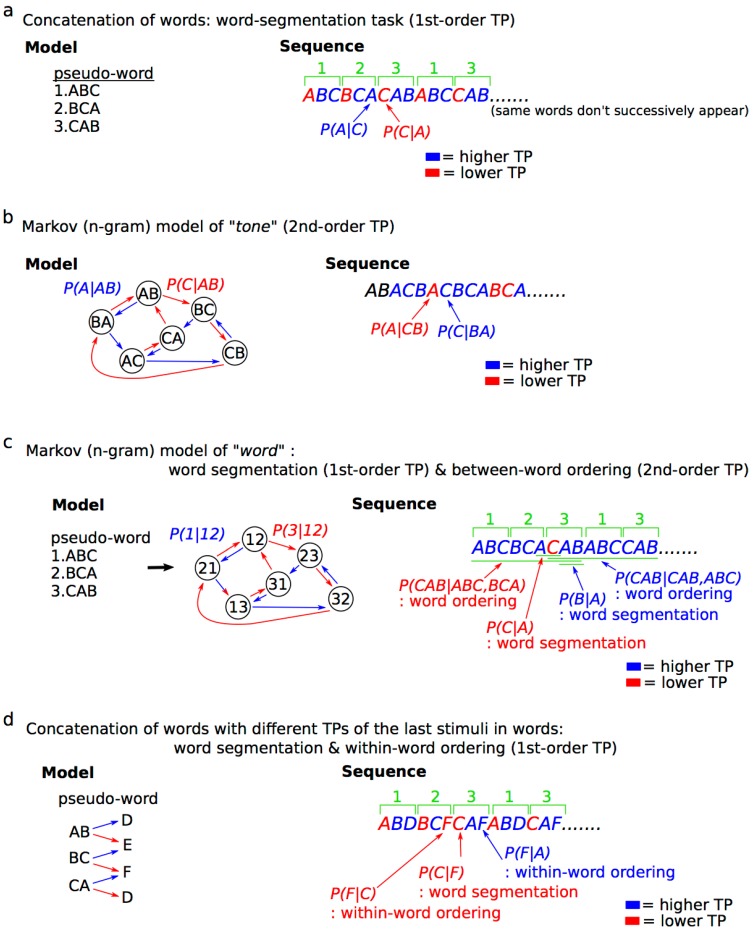
SL models and the sequences used in neural studies. All of the models and paradigms in sequences based on concatenation of words (**a**), Markov model of tone (**b**) and word (**c**), and concatenation of words with different TPs of the last stimuli in words (**d**) are simplified so that the characteristics of paradigms can be compared. In the example of word-segmentation paradigm (**a**), the same words do not successively appear. TP—transitional probability.

**Figure 3 brainsci-08-00114-f003:**
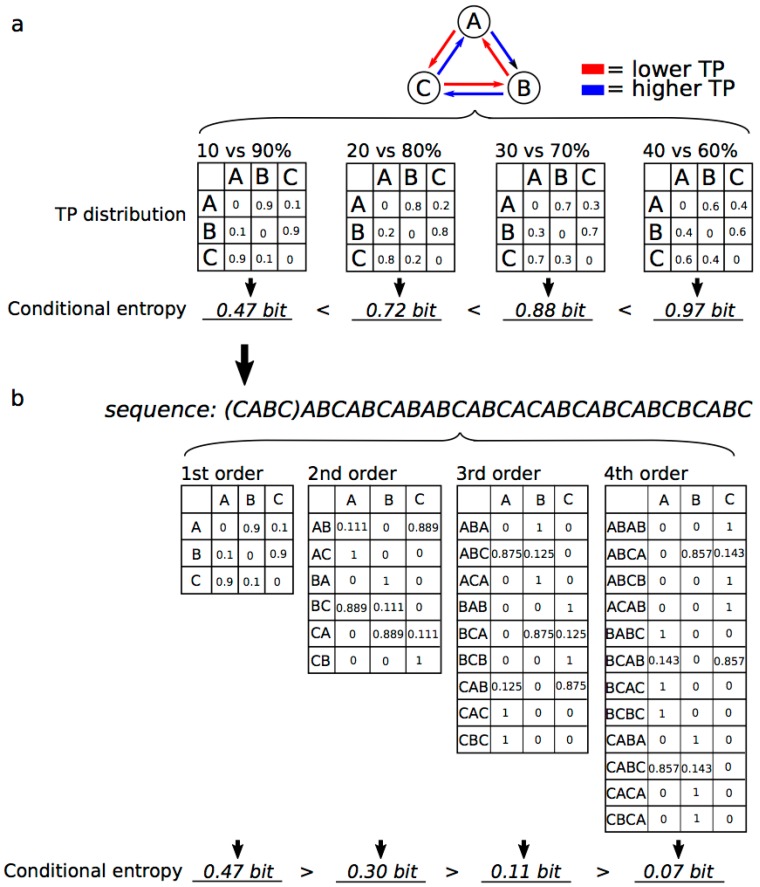
The entropy (uncertainty) of predictability in the framework of SL. The uncertainties depend on (**a**) TP ratios in a first-order Markov model (i.e., bigram model) and (**b**) orders of models in the TP ratio of 10% vs. 90%.

**Figure 4 brainsci-08-00114-f004:**
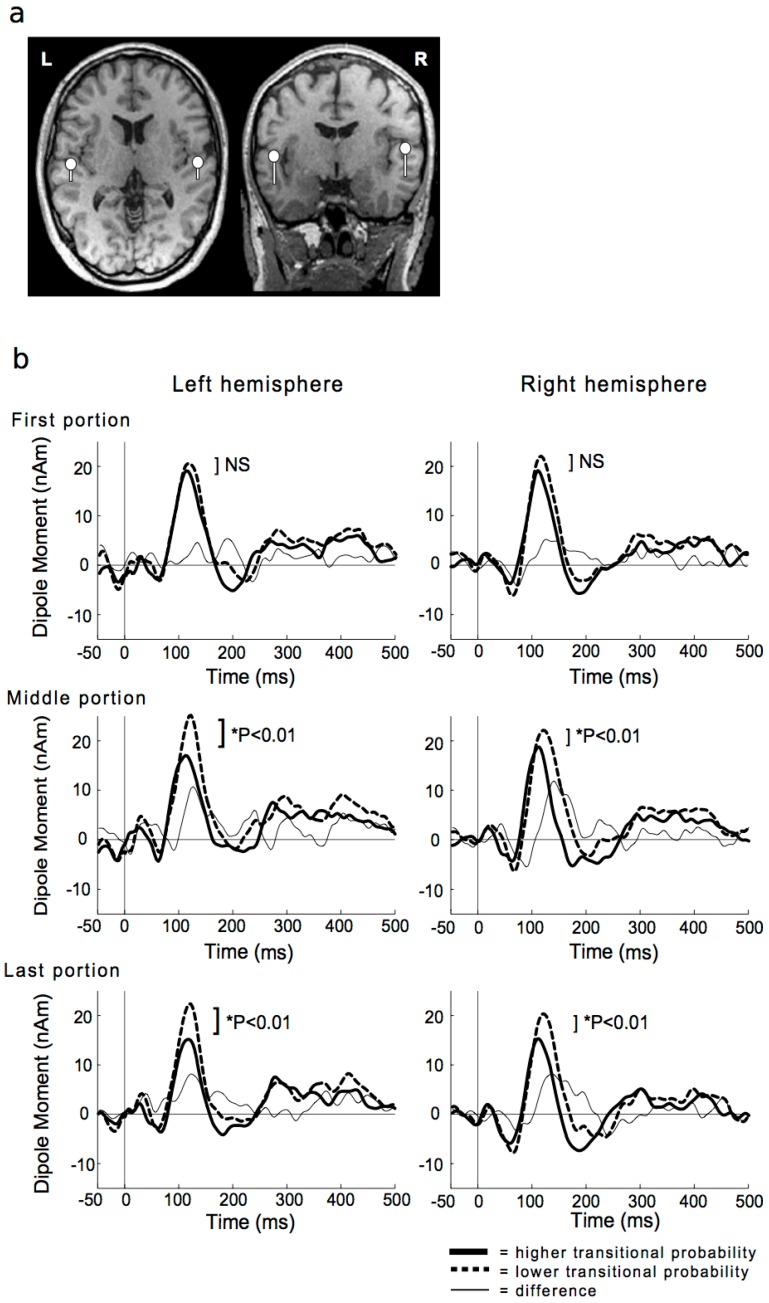
Representative equivalent current dipole (ECD) locations (dots) and orientations (bars) for the N100 m responses superimposed on the magnetic resonance images (**a**) (Daikoku et al., 2014 [32]; and the SL effects (**b**) (Daikoku et al., 2015 [10]) (NS = not significant). When the brain encodes the TP in a sequence, it expects a probable future stimulus with a high TP and inhibits the neural response to predictable stimuli. In the end, the SL effects manifest as a difference in amplitudes of neural responses to stimuli with lower and higher TPs (**b**).

**Table 1 brainsci-08-00114-t001:** Overview of neurophysiological correlations with auditory statistical learning. TP—transitional probability; ABR—auditory brainstem response; MMN—mismatch negativity; STS—superior temporal sulcus; STG—superior temporal gyrus; IFG—inferior frontal gyrus; PMC—premotor cortex; PTC—posterior temporal cortex.

Paradigms	Order of TP	Neural Correlates	References
Word segmentation	First-order	ABR	Skoe et al., 2015 [96]
		P50	Paraskevopoulos et al., 2012 [41]
		N100	Sanders et al., 2002 [94]
		MMN	Koelsch et al., 2016 [40] Moldwin et al., 2017 [98] Francois et a., 2017 [44]
		P200	De Diego Balaguer et al., 2007 [89] Francois et al., 2011 [46] Cunillera et al., 2006 [105]
		N200–250	Mandikal Vasuki et al., 2017 [47] Francois et al., 2017 [44]
		P300	Batterink et al., 2015 [83]
		N400	Cunillera et al., 2009 [88], 2006 [105] De Diego Balaguer et al., 2007 [89] Sanders et al., 2002 [94] Francois et al., 2011 [46]; 2013 [43]; 2014 [93]
		STS, STG	Farthouat et al., 2017 [91] Tremblay et al., 2012 [110] Paraskevopoulos et al., 2017 [45]
		Left IFG	Abla and Okanoya, 2008 [111] McNealy et al., 2006 [112] Paraskevopoulos et al., 2017 [45]
		PMC	Cunillera et al., 2009 [88]
		Hippocampus	Schapiro et al., 2014 [113]
Markov model	First-order	P50	Daikoku et al., 2016 [36]
		Wernicke’s area	Bischoff-Grethe et al., 2000 [114]
		Hippocampus	Harrison et al., 2006 [60]
	Higher-order	P50	Daikoku et al., 2017 [14]; 2017 [37]
		N100	Furl et al., 2011 [35] Daikoku et al., 2014 [32]; 2015 [10]; 2017 [14]
		P200	Furl et al., 2011 [35]
		Right PTC	Furl et al., 2011 [35]

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
