# Peer review of "Neurophysiological Markers of Statistical Learning in Music and Language: Hierarchy, Entropy and Uncertainty"

_brainsci, 2018, doi:10.3390/brainsci8060114_

Round 1

Reviewer 1 Report

Review of “Neurophysiological marker of statistical learning in music and language: hierarchy, entropy, and uncertainty”

The manuscript reviews recent literature on statistical learning (SL) and proposes that processes based on statistical learning may link music and language. It explains SL from the perspective of information theory, summarises the neural mechanisms underlying SL, and suggests possibilities of applying SL in therapy and pedagogy. The manuscript is well-organised and sheds light on an important connection between music and language. However, the manuscript would also benefit from significant editing to enhance the clarity of the arguments, especially for readers who are unfamiliar with statistical learning and relevant literature. More detailed comments follow. 

1.    Abstract. The writing needs work. Often these problems relate to an inappropriate choice of wording. Even the title is hard to digest and does not appear to convey the core argument of the paper. Then in the first sentence of the abstract, statistical learning is defined as a “method of learning” but also as an implicit, domain general “mechanism.” It would be better to break this first sentence into two - starting with a simple definition of statistical learning, and following with a new sentence about the implicit, domain general nature of SL. Similarly, the claim that “the information-theoretical aspect of SL can be represented in neurophysiological responses” will be confusing to most readers – all these ideas need to be considered more carefully, and revised accordingly. 

2.   Line 17-18. Readers will not understand the claim that a “range of works” (a) suggests overlapping and neural correlates of music and language; and (b) indicates impairment in humans with domain-specific learning disability”. The second claim seems to be a separate idea, so should perhaps be described in a separate sentence. 

3.   Lines 20. What does “mentally expressed” mean? Again, readability is hampered by imprecise wording. More care needs to be taken on the writing. 

4.   Lines 58. “BecauseSL is a domain-general learning system, [therefore] recent studies have detected the effects…” The logic of this sentence is unclear – sentence needs rethinking and revising.

5.   Lines 66-67. It is unclear why examining the relationships between the orders of TPs and entropy in SL is important. More elaboration is needed. This point also applies to the discussion section (see lines 396-397).

6.   Lines 84-87. For readers who are not in the area, the author needs to explain more clearly what nth-order Markov models and n-gram models are, and how such models are applied in the study of natural language and music. Additional text may be required. 

7.   Lines 120. Need to describe the word segmentation paradigm in much more detail, and explain more clearly how the paradigm is employed and in what context. Although stimuli associated with this paradigm is noted (“a concatenation of pseudo words”), there was no explanation of the actual procedure. 

8.   The argument being made in section 2.3 is not very clear or precise. The final sentence of that section seems very general, and also does not seem to follow from the preceding review. A more refined and valuable argument is needed.

9.   Line 141. Neural basis underlying … (tautological – “basis” already implies “underlying”). 

10.   Lines 149-153. N400 is described as the most robust ERP, but many other ERPs are listed and their connection to SL is unclear. The sentence implies virtually all ERPs are related to SL, but no insight is provided as to what they may reflect. Auditory responses at an early stage have been discussed in lines 154-162. Considerable clarification and refinement of the arguments are needed here. 

11.   Lines 163-175. It is unclear what argument is being made in this paragraph. Suggest that a clear argument is provided at the beginning of the paragraph, and why the point is relevant to the overall goal of the manuscript. 

12.Lines 184-185. “… brain connectivity between superior temporal sources and the left IFG is important for auditory SL.” Please clarify whether this refers to the bilateral superior temporal sources or the only left side. 

13.  Lines 251-252. “This suggests that musical training could contribute to improvement of language skills.” The evidence that is subsequently reviewed is related to auditory processing, but a more detailed argument should be made about how such changes might be associated with enhanced languageprocessing. It should also be emphasized in the paper that the findings are correlational, so do not by themselves indicate a causal connection. 

14.  Lines 358. The term “mental expression” is a little vague and might be confusing to some readers. Suggest using a simpler, more conventional phrase.  

15.  Lines 360-362. Wording is imprecise: language comprehension, decision-making and others mentioned here are not “mental representations” per se. Please refer to definitions of mental representation within cognitive science and use more precise wording. A similar issue arises on line 366.

16.  Lines 367-372. “A number of studies have reported, however, that humans cannot verbalise exactly what they have learned statistically, even when an SL effect is detected in neurophysiological responses [12,22,26-36]. Nevertheless, our previous study demonstrated a means of detecting the behavioural effects of SL by using an abstract medium such as a musical melody [22].” The point is unclear and needs to be fleshed out. The authors are implying that by using an abstract medium, one can behaviourally measure statistical learning, but not by non-abstract media. The point is peculiar and it is not clear what is meant by “abstract”. Clarification and elaboration are needed.

17.  Figure 2b and Figure 4 were not cited in the text.

18.  The manuscript could benefit from more detailed discussion of the specificity of SL as a point of convergence between music and language. Statistical learning can presumably be applied across any domain in which the conditional probabilities of sequential events vary systematically. 

END OF REVIEW

Author Response

I would be extremely grateful for the Reviewer’s insightful comments on our manuscript. I have studied the Reviewer’s comments very carefully and have made necessary corrections. I feel the comments have helped us significantly improve and refine the manuscript. The responses related to Reviewer’s comment are as follows.

Comment 1:

Abstract. The writing needs work. Often these problems relate to an inappropriate choice of wording. Even the title is hard to digest and does not appear to convey the core argument of the paper. Then in the first sentence of the abstract, statistical learning is defined as a “method of learning” but also as an implicit, domain general “mechanism.” It would be better to break this first sentence into two - starting with a simple definition of statistical learning, and following with a new sentence about the implicit, domain general nature of SL. Similarly, the claim that “the information-theoretical aspect of SL can be represented in neurophysiological responses” will be confusing to most readers – all these ideas need to be considered more carefully, and revised accordingly.

Response: I thank the Reviewer for this pertinent comment. As indicated by the reviewer, I revised the abstract.

Comment 2:

Line 17-18. Readers will not understand the claim that a “range of works” (a) suggests overlapping and neural correlates of music and language; and (b) indicates impairment in humans with domain-specific learning disability”. The second claim seems to be a separate idea, so should perhaps be described in a separate sentence.

Lines 20. What does “mentally expressed” mean? Again, readability is hampered by imprecise wording. More care needs to be taken on the writing.

Response: Thank you for the important indication. I revise them.

Comment 3:

Response: Thank you for the comment. I refined the Abstract section.

Comment 4:

Lines 58. “BecauseSL is a domain-general learning system, [therefore] recent studies have detected the effects…” The logic of this sentence is unclear – sentence needs rethinking and revising.

Response: Thank you for the comment. I revised the sentence.

Comment 5:

Lines 66-67. It is unclear why examining the relationships between the orders of TPs and entropy in SL is important. More elaboration is needed. This point also applies to the discussion section (see lines 396-397).

Response: I thank the Reviewer for this pertinent comment. Recent studies have also suggested that SL strategies in the brain depend on the hierarchy, order [12,27,50,51], entropy, and uncertainty in statistical structures [52]. Hasson et al. [70] also indicated that certain regions or networks perform specific computations of global or summary statistics (i.e., entropy) which are independent of local statistics (i.e., TP). I refined the points indicated by reviewer in Introduction and Conclusion Sections.

Comment 6:

Lines 84-87. For readers who are not in the area, the author needs to explain more clearly what nth-order Markov models and n-gram models are, and how such models are applied in the study of natural language and music. Additional text may be required.

Response: Thank you for the comment. The nth-order Markov and n-gram models were described in the section of 2.1.

Comment 7:

Lines 120. Need to describe the word segmentation paradigm in much more detail, and explain more clearly how the paradigm is employed and in what context. Although stimuli associated with this paradigm is noted (“a concatenation of pseudo words”), there was no explanation of the actual procedure.

Response: I thank the Reviewer for this pertinent comment. As indicated by the reviewer, I described he word segmentation paradigm in much more detail in the section of 2.3.

Comment 8:

The argument being made in section 2.3 is not very clear or precise. The final sentence of that section seems very general, and also does not seem to follow from the preceding review. A more refined and valuable argument is needed.

Response: I thank the Reviewer for this pertinent comment. As indicated by the reviewer, I revised the section of 2.3.

Comment 9:

Line 141. Neural basis underlying … (tautological – “basis” already implies “underlying”).

Response: I reworded it.

Comment 10:

Lines 149-153. N400 is described as the most robust ERP, but many other ERPs are listed and their connection to SL is unclear. The sentence implies virtually all ERPs are related to SL, but no insight is provided as to what they may reflect. Auditory responses at an early stage have been discussed in lines 154-162. Considerable clarification and refinement of the arguments are needed here.

Response: I thank the Reviewer for this pertinent comment. As indicated by the reviewer, I refined the section of 3.1.

Comment 11:

Lines 163-175. It is unclear what argument is being made in this paragraph. Suggest that a clear argument is provided at the beginning of the paragraph, and why the point is relevant to the overall goal of the manuscript.

Response: Thank you for the important indication. As indicated by the reviewer, I revise the paragraph.

Comment 12:

Lines 184-185. “… brain connectivity between superior temporal sources and the left IFG is important for auditory SL.” Please clarify whether this refers to the bilateral superior temporal sources or the only left side.

Response: Thank you for the comment. I revised it.

Comment 13:

Lines 251-252. “This suggests that musical training could contribute to improvement of language skills.” The evidence that is subsequently reviewed is related to auditory processing, but a more detailed argument should be made about how such changes might be associated with enhanced languageprocessing. It should also be emphasized in the paper that the findings are correlational, so do not by themselves indicate a causal connection.

Response: I thank the Reviewer for this pertinent comment. Because I discussed the relationships among SL, language processing, and music training latter in this section, I excluded the sentence.

Comment 14:

Lines 358. The term “mental expression” is a little vague and might be confusing to some readers. Suggest using a simpler, more conventional phrase. 

Response: Thank you for the comment. I revised it.

Comment 15:

Lines 360-362. Wording is imprecise: language comprehension, decision-making and others mentioned here are not “mental representations” per se. Please refer to definitions of mental representation within cognitive science and use more precise wording. A similar issue arises on line 366.

Response: Thank you for the comment. I revised it.

Comment 16:

Lines 367-372. “A number of studies have reported, however, that humans cannot verbalise exactly what they have learned statistically, even when an SL effect is detected in neurophysiological responses [12,22,26-36]. Nevertheless, our previous study demonstrated a means of detecting the behavioural effects of SL by using an abstract medium such as a musical melody [22].” The point is unclear and needs to be fleshed out. The authors are implying that by using an abstract medium, one can behaviourally measure statistical learning, but not by non-abstract media. The point is peculiar and it is not clear what is meant by “abstract”. Clarification and elaboration are needed.

Response: I thank the Reviewer for this pertinent comment. As indicated by the reviewer, I revised them.

Comment 17:

Figure 2b and Figure 4 were not cited in the text.

Response: Thank you for the indication. The Figure 2b and Figure 4 were cited in the section of 2.3. and 3.1.

Comment 18:

The manuscript could benefit from more detailed discussion of the specificity of SL as a point of convergence between music and language. Statistical learning can presumably be applied across any domain in which the conditional probabilities of sequential events vary systematically.

Response: Thank you for the indication. I described that point in the section of 2.3. and Discussion Section (5.3.).

Reviewer 2 Report

Dear Tatsuya Daikoku, i enjoyed reading your review about statistical learning in music and language. As i have nothing else to complain, what follows is a list of minor editing suggestions.

35-41: might be splitted into two sentences for easier comprehension
80:  "TP distribution" --> "TP distributions"
88 and following: in the final document, mathematical expressions in text should also be typesetted:  en+1 --> e_{n+1}  

95: although the figure is mostly self-explaining, a bit more textual explanation of the figure could help readers that first look at the figures before reading the text.
98: if possible, please give a short description for each subfigure (a) to (d).

please consider reordering of figure 3 and figure 2. figure 3 is mentioned before figure 2 in the text. so should be the ordering of figures, if the layout allows.

107, next line formula 2: the term  "(bits)" looks like beeing part of the formula - this is potentially confusing

137: unexpected, too high probability of word "Thus"

143: abbreviation ERP is not explained before. maybe a short introductory sentence with a reference might help readers coming from other research areas to understand that this paragraph is about EEG and brain potentials.

336: Figure 4 is not referenced in the text. Also, if space allows, briefly describe the SL effects in (b) in the figure caption.

405: first sentence is confusing.

Author Response

I wish to express our strong appreciation to the Reviewer for the insightful comments on our manuscript. I have studied the Reviewer’s comments very carefully and have made necessary corrections. I feel the comments have helped us significantly improve the manuscript.

Comment 1:

35-41: might be splitted into two sentences for easier comprehension

Response: Thank you for the comment. As indicated by the reviewer, this was splitted into two sentences.

Comment 2:

80:  "TP distribution" --> "TP distributions"

Response: I modified it.

Comment 3:

88 and following: in the final document, mathematical expressions in text should also be typesetted:  en+1 --> e_{n+1} 

Response: Thank you for the comment. I modified it.

Comment 4:

95: although the figure is mostly self-explaining, a bit more textual explanation of the figure could help readers that first look at the figures before reading the text.

98: if possible, please give a short description for each subfigure (a) to (d).

Response: I thank the Reviewer for this pertinent comment. I described them more details in Figure legends.

Comment 5:

please consider reordering of figure 3 and figure 2. figure 3 is mentioned before figure 2 in the text. so should be the ordering of figures, if the layout allows.

Response: Thank you for the comment. I modified it.

Comment 6:

107, next line formula 2: the term  "(bits)" looks like beeing part of the formula - this is potentially confusing

Response: I thank the Reviewer for this pertinent comment. As indicated by the reviewer, I excluded it.

Comment 7:

137: unexpected, too high probability of word "Thus"

Response: Thank you for this pertinent comment. As indicated by the reviewer, I excluded it.

Comment 8:

143: abbreviation ERP is not explained before. maybe a short introductory sentence with a reference might help readers coming from other research areas to understand that this paragraph is about EEG and brain potentials.

Response: The abbreviation ERP has been explained in the Introduction section, but I also introduced ERP in the Introduction.

Comment 9:

336: Figure 4 is not referenced in the text. Also, if space allows, briefly describe the SL effects in (b) in the figure caption.

Response: I thank the Reviewer for this pertinent comment. Figure 4 was not referenced in the section of 3.1. In addition, I describe the SL effects in (b) in the figure caption.

Comment 10:

405: first sentence is confusing.

Response:

Thank you for indication. I revised it.

Reviewer 3 Report

This 
paper reviews a large body of work on the neural correlates of Statistical Learning (SL) in music and language. 

Overall the paper is interesting and can provide a valuable contribution to a special issue on the "Advances in the Neurocognition of Music and Language". However, I have some concerns regarding the framing of the manuscript and several parts of the manuscript need clarification. Additionally, I'm of the opinion that the paper can have a significant contribution to the field but in order to do so it needs to go a step further than simply reviewing / listing previous studies (I give some concrete suggestions below).

Major comments:

- Throughout the manuscript it is not clear whether "domain" in "domain-general learning system" (first use line 36) refers to sensory modality or different faculties in cognition.  Given the current debate about domain-specificity vs. generality I was surprised to see domain-generality stated as a fact. 

-       SL is presented as “a process by which the brain automatically calculates the TPs of …” (e.g., line 37). Whereas conditional probabilities are indeed the statistical information present in the input the sensitivity to these statistical regularities could be not the result of statistical computations on individual elements, but a by-product of chunking (as Perruchet has argued in many papers). This alternative approach should at least be acknowledged or argued against.

-       The paper stresses the distinction between local and global statistics (with the latter defined as entropy). This point would become a lot clearer if it was linked to / contrasted with the other SL literature making the distinction between learning transitional and distributional statistics (e.g., Thiessen, Kronstein, Hufnagle, 2013).

 -       The motivation for an entropy/uncertainty perspective becomes somewhat clear only in the general discussion. It would help the reader to state a clear motivation before the current line 73. Also, I miss a discussion of studies that have taken this approach… How have they contributed to linking different disciplines studying SL?

 -       Figure 2 needs further clarification in the caption or the text. The titles for a-d are not informative enough. Some questions to be addressed:

o   Is “word segmentation” used for the language domain only or as a name of the paradigm independent of the stimuli? I would argue the typical paradigm is better represented by ABCDEFGHIABC… (the same comment holds for Figure 3)

o   In 2c not sure I follow this to 5th order?

o   In 2d it is not clear what the within-word ordering refers to. 

-       Lines 145-147 need unpacking.

-       The paper provides a good overview of the neurophysiological correlates of auditory SL but discusses neither the meta-knowledge we get from a table like the one provided nor the potentially interesting discrepancies it reveals. For example, whereas the N400 seems the most robust index of word segmentation studies employing the 1st order Markov model (in Figure 2 proposed to capture exactly word segmentation) have not found it? In other words, what can we learn from Table 1?

-       Line 378-278. I’m not sure I agree regarding the individual differences statement. The statement should be motivated and the comparison to behavior should be more precise as some new behavioral tasks are reliably measuring individual differences (see e.g., Siegelman, Bogaerts, Christiansen & Frost, 2017)

Other minor comments

-       Title: A marker or markers?

-       p. 2, use subscript for n and n+1 consequently

-       80-82: a TP distribution or TP distributions

-       Line 197-199: “… are responsible for SL of word segmentation”: I doubt the author believes the N400 is responsible for SL

-       Please provide references for “the hypothesis that humans learn larger structures such as phrases first and… “ (line 219-222)

-       Line 275-277. It is not clear what is meant by familiar/novel language.

Author Response

I would be extremely grateful for the Reviewer’s insightful comments on our manuscript. I have studied the Reviewer’s comments very carefully and have made necessary corrections. I feel the comments have helped us significantly improve and refine the manuscript. The responses related to Reviewer’s comment are as follows.

Comment 1:

Response:

Throughout the manuscript it is not clear whether "domain" in "domain-general learning system" (first use line 36) refers to sensory modality or different faculties in cognition. Given the current debate about domain-specificity vs. generality I was surprised to see domain-generality stated as a fact.

Response: I thank the Reviewer for this pertinent comment. I was going to express that, although there are specific neural bases underlying language and music comprehension, many researchers also suggest that SL is a domain-general learning system that shared in cognition of music and language, in the sensory modalities such as auditory and visual, and in the species such as humans and monky, as reported in this paper. It doesn’t mean that music and language cognition itself are domain-general. I described it in the Introduction section.

Comment 2:

SL is presented as “a process by which the brain automatically calculates the TPs of …” (e.g., line 37). Whereas conditional probabilities are indeed the statistical information present in the input the sensitivity to these statistical regularities could be not the result of statistical computations on individual elements, but a by-product of chunking (as Perruchet has argued in many papers). This alternative approach should at least be acknowledged or argued against.

Response: I’m grateful for this pertinent comment. As indicated by the reviewer, I revised them.

Comment 3:

The paper stresses the distinction between local and global statistics (with the latter defined as entropy). This point would become a lot clearer if it was linked to / contrasted with the other SL literature making the distinction between learning transitional and distributional statistics (e.g., Thiessen, Kronstein, Hufnagle, 2013).

Response: I thank the Reviewer for this pertinent comment. As indicated by the reviewer, it was contrasted SL literature making the distinction between learning transitional and distributional statistics in the Introduction and discussion section [5.1.].

Comment 4:

The motivation for an entropy/uncertainty perspective becomes somewhat clear only in the general discussion. It would help the reader to state a clear motivation before the current line 73. Also, I miss a discussion of studies that have taken this approach… How have they contributed to linking different disciplines studying SL?

Response: I thank the Reviewer for this pertinent comment. As indicated by the reviewer, I described the motivation for an entropy/uncertainty perspective in the Introduction section. Furthermore, I described how an entropy/uncertainty perspective has contributed to linking different disciplines studying SL in the Discussion section.

Comment 5:

Figure 2 needs further clarification in the caption or the text. The titles for a-d are not informative enough. Some questions to be addressed:

Response: I thank the Reviewer for this pertinent comment. As indicated by the reviewer, the titles were revised.

Comment 6:

Is “word segmentation” used for the language domain only or as a name of the paradigm independent of the stimuli? I would argue the typical paradigm is better represented by ABCDEFGHIABC… (the same comment holds for Figure 3)

Response: I thank the Reviewer for this pertinent comment. The word segmentation is used as a name of the paradigm independent of the stimuli. I used just A, B, and C to simplify the example of paradigm and to link to Figure 3. In figure 3, if further events (e.g., D,E,F,G,H,I) are used, TP matrices become complicated very much (9x9 matrices), and difficult to understand.

Comment 7:

In 2c not sure I follow this to 5th order?

Response: I thank the Reviewer for this pertinent comment. As indicated by the reviewer, I revised it.

Comment 8:

In 2d it is not clear what the within-word ordering refers to.

Response: Thank you for the helpful indication. I improved them.

Comment 9:

Lines 145-147 need unpacking.

Response: I revised them.

Comment 10:

The paper provides a good overview of the neurophysiological correlates of auditory SL but discusses neither the meta-knowledge we get from a table like the one provided nor the potentially interesting discrepancies it reveals. For example, whereas the N400 seems the most robust index of word segmentation studies employing the 1st order Markov model (in Figure 2 proposed to capture exactly word segmentation) have not found it? In other words, what can we learn from Table 1?

Response: I thank the Reviewer for the important indication. At least, SL effects of ABR and P50 may suggest top-down prediction in SL. As for N400 in word segmentation, I hypothesize that phenomenon of word segmentation itself may be related to not only statistical computation but also chunking or language-specific function. At least, in our previous studies using Markov models, paradigm itself is not for detecting N400 (SOA is short). Future study will be needed to verify SL effects of N400 using Markov model. I described them in the section of 3.1.

Comment 11:

Line 378-278. I’m not sure I agree regarding the individual differences statement. The statement should be motivated and the comparison to behavior should be more precise as some new behavioral tasks are reliably measuring individual differences (see e.g., Siegelman, Bogaerts, Christiansen & Frost, 2017)

Response: I thank the Reviewer for this pertinent comment. As indicated by the reviewer, I revised them.

Comment 12:

Title: A marker or markers?

Response: As indicated by the reviewer, there are some markers. I modified it.

Comment 13:

p. 2, use subscript for n and n+1 consequently

Response: Thank you for the comment. I revised them.

Comment 14:

80-82: a TP distribution or TP distributions

Response: I modified it.

Comment 15:

Line 197-199: “… are responsible for SL of word segmentation”: I doubt the author believes the N400 is responsible for SL

Response: I thank the Reviewer for this pertinent comment. As indicated by the reviewer, I revised them.

Comment 16:

Please provide references for “the hypothesis that humans learn larger structures such as phrases first and… “ (line 219-222)

Response: Thank you for the comment. I revised the sentence more correctly.

Comment 17:

Line 275-277. It is not clear what is meant by familiar/novel language.

Response: Thank you for the comment. I described them in more detail.

Round 2

Reviewer 1 Report

The revisions have improved the quality of this manuscript, which will make an excellent contribution to the special issue. I have only a few further requests. 

Please check for minor problems associated with use of English as a second language. For example, "A number of researches ..." - the writing is generally excellent but the manuscript should be checked carefully by the author.

A more detailed introduction and tutorial is needed on the unified brain theory. This introduction should outline what UBT offers to advance our understanding of the brain, and how it helps to fill in gaps of former theoretical frameworks for understanding the brain. It may also help to provide an assessment of how influential this theory has been, and whether it has become integrated into mainstream research, or whether it is still developing. I feel that a few sentences would not be sufficient. Rather, a "tutorial" on unified brain theory would take at least an additional paragraph. However, this could be easily drafted by drawing upon some of the basic introductory points made in Friston's 2010 review article. 

The manuscript, as written, seems very uncritical of SL as a way of understanding links between music and language. Readers will want to know which properties of music and language - and which points of convergence - cannot be readily explained by SL and unified brain theory. These limitations could be placed in a final section of the paper entitled "Challenges and future prospects" or something similar. 

Author Response

RESPONSE TO REVIEWER 1:

I would be extremely grateful for the Reviewer’s insightful comments on our manuscript. I have studied the Reviewer’s comments very carefully and have made necessary corrections. I feel the comments have helped us significantly improve and refine the manuscript. The responses related to Reviewer’s comment are as follows.

Comment 1:

Please check for minor problems associated with use of English as a second language. For example, "A number of researches ..." - the writing is generally excellent but the manuscript should be checked carefully by the author.

Response: I thank the Reviewer for this pertinent comment. I revised them.

Comment 2:

A more detailed introduction and tutorial is needed on the unified brain theory. This introduction should outline what UBT offers to advance our understanding of the brain, and how it helps to fill in gaps of former theoretical frameworks for understanding the brain. It may also help to provide an assessment of how influential this theory has been, and whether it has become integrated into mainstream research, or whether it is still developing. I feel that a few sentences would not be sufficient. Rather, a "tutorial" on unified brain theory would take at least an additional paragraph. However, this could be easily drafted by drawing upon some of the basic introductory points made in Friston's 2010 review article.

Response: Thank you very much for the indication. As indicated by the reviewer, I described about UBT in more details.

Comment 3:

The manuscript, as written, seems very uncritical of SL as a way of understanding links between music and language. Readers will want to know which properties of music and language - and which points of convergence - cannot be readily explained by SL and unified brain theory. These limitations could be placed in a final section of the paper entitled "Challenges and future prospects" or something similar.

Response: I thank the Reviewer for this pertinent comment. I added a section on it and described them.